# Inter-Segmental Coordination During Soccer Instep Kicking: A Vector-Coding Comparison Between Experienced Athletes and Novices

**DOI:** 10.3390/bioengineering12111151

**Published:** 2025-10-24

**Authors:** Liwen Zhang, Meizhen Zhang, Hui Liu

**Affiliations:** 1Biomechanics Laboratory, School of Physical Education and Health Engineering, Taiyuan University of Technology, Taiyuan 030024, China; zhangliwen1399@163.com (L.Z.); meizhen1116@163.com (M.Z.); 2Biomechanics Laboratory, College of Human Movement Science, Beijing Sport University, Beijing 100084, China

**Keywords:** motor control, dynamic system theory, coupling angle, coordination, kicking performance improvement

## Abstract

The purpose of this study was to characterize the inter-segmental coordination of hip, knee, and ankle movement of the kicking leg during instep kicking for experienced athletes and novices, using vector coding as a non-linear technique. Motion capture and electromyographic data were collected for 14 soccer-majored college students and 32 novices performing the instep kicking task. The percentage of time spent on the coordination patterns, defined based on hip–knee and knee–ankle coupling angles, was calculated and compared. The agonist–antagonist activity ratio was calculated and compared. The time percentages of the knee–ankle shank dominance of the experienced athletes during the whole kicking movement were significantly greater than those of the novices (*p* < 0.050). Athletes achieving greater maximum ball speed had more knee flexion dominant coordination patterns in the back swing and leg-cocking, and knee extension dominant coordination patterns in the leg acceleration phase. The lower activity ratio of the tibialis anterior and gastrocnemius muscles contributed significantly to increasing kicking accuracy. These results underscore the value of vector coding in identifying key inter-segmental coordination features and directly support targeted soccer kick training. The dynamic stability exercises involving knee flexion and extension to optimize power transfer for speed, as well as activation and relaxation control exercises of the lower leg muscles to improve the kicking accuracy, may be effective ways to enhance instep kicking motor control ability and performance for soccer athletes.

## 1. Introduction

Inter-segmental coordination is one of the most important factors affecting sports performance. From a dynamical systems perspective of motor control, inter-segmental coordination has been defined as the mastery of redundant degrees of freedom of the musculoskeletal system employed during sport movements to produce a controllable system [1]. The two most common non-linear techniques employed by dynamical system theorists for quantifying inter-segmental coordination appear to be vector coding and continuous relative phase (CRP) [2]. Compared to CRP, vector coding provides an additional insight into the dominance of one segment over another, and this can offer more valuable information in a clinical setting [3,4]. In vector coding, the coupling angle was calculated. The time spent in each inter-segmental coordination pattern, defined based on the value of the coupling angle, was usually used for further statistics. These patterns include in-phase (where both segments rotate in the same direction), anti-phase (where segments rotate in opposite directions), and the intermediate patterns of proximal and distal dominance (where movement is dominated by the upper or lower segment, respectively). The specific categorization followed the standard quadrants established in vector coding analysis [3,5]. In addition, the relative activity between agonist and antagonist muscles, expressed as the ratio of the two muscles, further describes the coordinated movement of the muscles during a specific sport movement [6]. Most previous investigations have reported that high-level players achieving better sports performance may exhibit better inter-segmental coordination patterns during movements [7,8,9,10,11,12]. For example, Chardonnens et al. [7] found that the players achieving longer jumps had lower shank-thigh mean CRP values (r = –0.41), indicating a greater overall lead of the thighs over the shanks, compared to the athletes with shorter jumps. The understanding of inter-segmental coordination for specific movements is necessary for providing insights regarding the motor control mechanisms and facilitating the improvement of sports performance.

To effectively achieve a high ball speed and improve the chances of scoring in the sport of soccer, identifying the inter-segmental coordination patterns for instep kicking is critical. Instep kicking is one of the most fundamental and frequently used skills in soccer. The speed of the ball and the kicking accuracy are the two main biomechanical indicators of the instep kicking success, and they depend on various factors [13]. Although the majority of previous studies on soccer kick biomechanics have demonstrated that the ball speed and kicking accuracy were significantly correlated with the hip, knee, and ankle kinematics of the kicking leg before the foot contact with the ball [14,15], a recent study found that a combination of greater joint angle of hip extension and knee flexion of the kicking limb correlated with better kicking performance [16]. These results suggested that more attention should be focused on the biomechanics of multiple segments of the kicking leg since the success of the instep kicking is attributed to the precise control of the joints and muscles.

Although significant efforts have been made in the last three decades to identify the inter-segmental coordination for instep kicking, the relationship between the inter-segmental coordination patterns and kicking performance remains unclear. A possible explanation is that there is a lack of quantitative research in the literature. Several studies have established the classical inter-segmental coordination pattern of instep kicking as a proximal-to-distal sequence of segmental movements in the kicking leg, facilitating effective energy transfer from the hip to the foot [13,17,18,19]. Studies utilizing angle–angle plots have further elucidated differences in this inter-segmental coordination pattern between skill levels. These studies suggest that the pattern is refined, characterized by a more precise temporal sequencing and a shift in the relative contribution between proximal and distal segments for experienced soccer players [11,18], or for novice players after a period of time of practice [10,20]. Furthermore, quantitative evidence from a study using CRP analysis reinforces this, demonstrating distinct coordination patterns between elite and amateur players and, importantly, directly linking these patterns to foot speed [21]. Although a recent study reported that the dominant coordination patterns for the hip and knee movement of the kicking leg are shank dominance and in-phase during the back swing and the forward swing phase during instep kicking, respectively [22], it did not quantitatively investigate how these coordination patterns influence kicking performance.

The purpose of this study was to characterize the inter-segmental coordination of the hip, knee, and ankle movements of the kicking leg during instep kicking for experienced athletes and novices. We hypothesized that there would be a significant difference in the time spent percentage of each of the hip–knee and knee–ankle inter-segmental coordination patterns, and the activity ratio of the agonist and antagonist muscles of the kicking leg between experienced athletes and novices. We also hypothesized that the maximum ball speed and the kicking accuracy would be significantly correlated with the time spent percentage of each of the hip–knee and knee–ankle inter-segmental coordination patterns, and the activity ratio of the agonist and antagonist muscles of the kicking leg for the experienced athletes.

## 2. Materials and Methods

### 2.1. Participants

Thirty-two soccer novice college students (age = 21.4 ± 1.9 years, height = 178.0 ± 6.5 cm, and body mass = 70.9 ± 9.4 kg) and fourteen male soccer-majored college students (age = 21.9 ± 2.2 years, height = 178.7 ± 4.5 cm, and body mass = 73.8 ± 10.3 kg) volunteered to participate in this study. The novices had the experience of instep kicking through self-practice but had never participated in any organized soccer games or received any formal soccer training. The soccer-major students were certified as “Level Two Athletes” under the Chinese national ranking system, denoting a nationally competitive level of performance. The participants were right-foot dominant without known lower extremity injuries six months before the study. The use of human subjects was approved by the Institutional Review Board of Beijing Sport University. Written consent was obtained from each participant before any data collection.

### 2.2. Protocols

Each participant had 10 min of warm-up, including running, stretching exercises, and several practice kicking exercises. Following warm-up, reflective markers were attached bilaterally to the anterior superior iliac spine (ASIS), mid-thigh, medial and lateral femoral condyles, tibial tuberosity, medial and lateral malleoli, posterior calcaneus, and first and fifth metatarsal heads. An additional marker was attached between lumbar vertebras 4 and 5 (L4–5). A reflective marker was also attached to the soccer ball for estimating the time of foot–ball contact. Surface electromyographic (EMG) electrodes were placed over the muscle belly of the rectus femoris, lateral femoris, medial femoris, semimembranosus, biceps femoris, tibialis anterior, and lateral gastrocnemius muscle of the dominant leg. A maximal voluntary contraction (MVC) was performed to collect MVC EMG data.

Each participant then performed ten successful trials for the instep kicking tasks, with a one-minute rest between two consecutive trials. Particularly, the participant was instructed to have an approach of two steps angled at 45° to the kicking direction, and to kick the stationary ball on the ground with the instep portion of the foot as hard as possible. Considering the accuracy demand, the participant was also instructed to kick the ball against the center of the goal (3 height × 2 m width) located five meters from the ball as hard as possible [21].

### 2.3. Data Collection

The 3-D coordinates of the reflective markers in each kicking trial were recorded using a motion capture data collection system (Raptor-4; Motion Analysis, Santa Rosa, CA, USA) comprising 8 cameras surrounding the capture volume, with a sampling rate of 200 frames/s. EMG data were recorded using a Delsys Trigno™ Wireless EMG acquisition system (Delsys Inc., Natick, MA, USA) at a sample rate of 2000 samples/channel/s. The motion capture and EMG data collections were time synchronized using the Delsys Triger Synchronizer (Delsys Inc., Natick, MA, USA).

The 2-D coordinates of the ball trajectory and the position where the ball enters the goal were recorded using two digital video cameras (GC-PX100, JVC, Tokyo, Japan) with a resolution of 1920 × 1080 at a sampling frequency of 60 frames/s, respectively [22,23]. One camera was set up five meters on the right side of the ball and perpendicular to the sagittal plane of the kicking movement, while another camera was set up behind the goal and parallel to the sagittal plane of the kicking movement.

### 2.4. Data Reduction

The kicking movement was divided into three phases [24]. The back swing phase begins with the toe-off of the kicking leg and ends with maximum hip extension of the kicking leg. The leg-cocking phase begins with maximum hip extension of the kicking leg and ends with maximum knee flexion of the kicking leg. The leg acceleration phase begins with maximum knee flexion of the kicking leg and ends with contact of the kicking foot with the ball [24].

The raw 3-D trajectories of all but one reflective marker were filtered using a Butterworth low-pass filter at an estimated optimum frequency of 13 Hz [25,26,27]. The raw 3-D trajectories of the single marker on the ball were not filtered, as any filtering of this trajectory created ambiguity in the identification of the specific frame when ball impact occurred [24].

The 3-D coordinates of the hip, knee, and ankle joint center were calculated from the 3-D coordinates of the reflective markers and anatomical data as previously described [28,29]. The pelvis segment reference frame was determined using the 3D coordinates of the left and right ASISs and the L4–5 joint. The thigh reference frame was determined using the 3-D coordinates of the hip and knee joint centers, and the markers on the medial and lateral tibial condyles. The lower leg reference frame was determined using the 3-D coordinates of the knee and ankle joint centers, and the markers on the medial and lateral tibial condyles. The foot reference frame was determined using the 3-D coordinates of the ankle joint center, posterior calcaneus, and first and fifth metatarsal heads [27,30]. The hip, knee, and ankle Cardan joint angles between adjacent segment reference frames were calculated in order of flexion–extension, valgus–varus, and internal rotation [27,30], and were normalized and time-scaled to 100% of the kicking movement.

The vector coding method was used to quantify the hip–knee and knee–ankle coordination patterns of the kicking leg over the time course of kicking [3,22]. The coupling angle (γ) was defined as the angle of the vector between the two adjacent data points in time on the angle–angle plot relative to the right horizontal and was calculated as$$\gamma_{j,i}=\tan^{-1}\left(\frac{y_{j,i+1-}y_{j,i}}{x_{j,i+1-}x_{j,i}}\right)$$
where 0° ≤ γ ≤ 360°, and i is a data point of the j_th_ trial. Mean coupling angles (${\bar{\gamma }}_{i}$) were computed using circular statistics since these coupling angles are directional in nature [31]. Within a subject and then across the group, ${\bar{\gamma }}_{i}$ was calculated from the mean horizontal (${\bar{x}}_{i}$) and vertical (${\bar{y}}_{i}$) components at each data point:$${\bar{x}}_{i}=\;\frac{1}{n}\sum_{j=1}^{n}\left(\cos y_{j,i}\right)$$$${\bar{y}}_{i}=\frac{1}{n}\sum_{j=1}^{n}\left(\sin y_{j,i}\right)$$$${\bar{\gamma }}_{i}=\left\{\begin{array}{c} \arctan\left({\bar{y}}_{i}/{\bar{x}}_{i}\right),\;if\;{\bar{x}}_{i}>0\\ 180+arctan({\bar{y}}_{i}/{\bar{x}}_{i}),\;if\;{\bar{x}}_{i}<0 \end{array}\right.$$

Based on the mean coupling angle, the coordination patterns were defined (Table 1), including in-phase, anti-phase, proximal dominance, and distal dominance. The patterns are found at the vertical, horizontal, and 45° diagonals. For the hip–knee coordination, a coupling angle of 45° or 225° (a positive diagonal) indicates an in-phase pattern, where the two segments rotate in the same direction, for example, hip flexion countered by knee extension. Conversely, coupling angles of 135° or 315° (a negative diagonal) correspond to an anti-phase pattern, where the two segments rotate in opposite directions, for example, concurrent hip and knee flexion. When coupling angles parallel the horizontal (γ = 0° or 180°), there is a thigh dominance, where there is hip rotation but no knee rotation. Vertically directed coupling angles (γ = 90° or 270°) indicate a shank dominance, where the knee segment rotates exclusively [3]. When coupling angles deviate from these respective vertical, horizontal, and diagonal vectors, the movement patterns are less pure.

The time spent percentage of each coordination pattern for the hip, knee, and ankle movement of the kicking leg was calculated.

The raw EMG signals were filtered using a band-pass digital filter with a 10 Hz high-pass cutoff and an 800 Hz low-pass cutoff, and then rectified [27]. This 800 Hz low-pass cutoff was selected to preserve the full spectral content of the signal, as the rapid, ballistic nature of the instep kick may elicit neuromuscular activation across a broader frequency spectrum. This approach is directly aligned with methodologies established in studies analyzing similar explosive actions, such as the instep kick [27] and high-speed running [32]. The band-pass filtered and rectified EMGs were filtered using a low-pass digital filter again at a cutoff frequency of 20 Hz to obtain the linear envelop EMGs [27]. Subsequently, the EMG amplitude for each muscle was normalized to its corresponding MVC value obtained during the pre-test maximal contractions. The activity ratio of the agonist and antagonist muscles was calculated as the ratio of the envelop EMGs of the quadriceps femoris and hamstrings, and the tibialis anterior and gastrocnemius muscles, respectively [6]. The average activity ratio of the agonist and antagonist muscles was then calculated for each of the kicking tasks.

The translational velocity of the ball was derived from 2D video analysis because the single-marker 3D setup could not accurately track the center of mass of the rotating ball [14]. The 2-D coordinates of the ball center in the video were obtained using Fastmove 3D Motion (FastMove Inc., Dalian, China). The ball speed was defined as the ratio of the displacement of the ball to the time between every two adjacent frames. The maximum ball speed is the maximum value of the ball speed during the kicking movement. The kicking accuracy was defined as the distance between the ball center and the goal center at the moment when the ball enters the goal, which was obtained by the video shooting. While it is acknowledged that kicking accuracy in a real-game context is influenced by additional factors (such as the goalkeeper’s positioning and specific tactical aims), this metric was chosen for its objectivity, reliability, and widespread use in biomechanical studies [33].

### 2.5. Data Analysis

The independent sample *t*-test was used to examine the differences in maximum ball speed, distance between the ball center and goal center, the time spent percentage of each of the hip–knee and knee–ankle coordination patterns, and the activity ratio of the quadriceps femoris and hamstring, and the tibialis anterior and gastrocnemius muscles during the instep kicking tasks among experienced athletes and novices. Effect sizes for group comparisons were quantified using Cohen’s d for independent sample *t*-tests.

A Pearson correlation (i.e., correlation coefficient r and the associated *p*-value) was used to evaluate the linear relationships between the maximum ball speed and distance between the ball center and goal center, with the time spent percentage of each of the hip–knee and knee–ankle coordination patterns, and the activity ratio of the quadriceps femoris and hamstring, and the tibialis anterior and gastrocnemius muscles of the experienced athletes.

A Type I error rate greater than or equal to 0.05 was chosen as the indication of statistical significance.

## 3. Results

The independent sample *t*-test revealed that the maximum ball speed of the experienced athletes was significantly greater than that of the novices, while the distance between ball center and goal center of the experienced athletes was significantly smaller than that of the novices (Table 2).

For both experienced athletes and novices, the hip joint was in extension motion in the back swing phase and then in flexion motion in the leg-cocking and leg acceleration phase (Figure 1A). The knee joint was in extension motion at the beginning portion of the back swing phase, and then in flexion motion until the moment of maximum knee flexion, and finally in extension motion in the leg acceleration phase (Figure 1B). For both experienced athletes and novices, the ankle joint was in dorsiflexion motion in the back swing phase (Figure 1C). The ankle joint was in dorsiflexion motion at the beginning portion of the leg-cocking phase, in plantar flexion motion before and after the moment of maximal knee flexion, and then in dorsiflexion motion in the later portion of the leg acceleration phase for the experienced athletes (Figure 1C). The ankle was in plantar flexion motion in the leg-cocking and early portion of the leg acceleration phase, and then in dorsiflexion motion in the later portion of the leg acceleration phase, for the novices (Figure 1C).

The experienced athletes and novices had similar hip–knee and knee–ankle coupling angle curves (Figure 2A,B). The dominant hip–knee coordination pattern was shank dominance and in-phase in the back swing, anti-phase and shank dominance in the leg-cocking, and shank dominance and in-phase in the leg acceleration phase during the kicking movement (Figure 2A, Table 3). The dominant knee–ankle coordination pattern was shank dominance during the whole kicking movement (Figure 2B, Table 4).

The time spent percentage of the hip–knee in-phase and anti-phase coordination patterns in the back swing phase of the experienced athletes was significantly greater than that of the novices. However, the time spent percentage of the hip–knee thigh dominance in the back swing, and the in-phase coordination pattern in the leg-cocking phase of the experienced athletes was significantly smaller than that of the novices. No significant difference in the time spent percentage of other hip–knee coordination patterns was detected between the two groups of participants (Table 3).

The time spent percentage of the knee–ankle shank dominance in the three kicking phases of the experienced athletes was significantly greater than that of the novices. However, the time spent percentage of the knee–ankle anti-phase in the leg-cocking and leg acceleration phase, and the foot dominance in the leg-cocking phase of the experienced athletes was significantly smaller than that of the novices. No significant difference in the time spent percentage of other knee–ankle coordination patterns was detected between the two groups of participants (Table 4).

No significant difference in the activity ratio of the quadriceps femoris and hamstring muscles was detected between the two groups of participants. The activity ratio of the tibialis anterior and gastrocnemius muscles of the experienced athletes was significantly smaller than that of the novices (Table 5).

The time spent percentages of hip–knee in-phase in the back swing, hip–knee shank dominance, and knee–ankle foot dominance in the leg acceleration phase were significantly positively correlated with the maximum ball speed, while the time spent percentages of hip–knee thigh dominance, knee–ankle foot dominance in the back swing, knee–ankle anti-phase in the leg-cocking phase, hip–knee in-phase, and thigh dominance in the leg acceleration phase were significantly negatively correlated with the maximum ball speed. Other time spent percentages of hip–knee and knee–ankle coordination patterns had no significant correlations with maximum ball speed (Figure 3 and Figure 4).

The time spent percentages of hip–knee thigh dominance and knee–ankle foot dominance in the back swing were significantly positively correlated with the distance between ball center and goal center, while other time spent percentages of hip–knee and knee–ankle coordination patterns had no significant correlations with the distance between ball center and goal center (Figure 5 and Figure 6).

No significant correlations were found between the activity ratio of the quadriceps femoris and hamstring, the tibialis anterior and gastrocnemius muscles, and the maximum ball speed (Figure 7). The activity ratio of the tibialis anterior and gastrocnemius muscles was significantly positively correlated with the distance between ball center and goal center, while no significant correlations were found between the activity ratio of the quadriceps femoris and hamstring muscles and the distance between ball center and goal center (Figure 8).

## 4. Discussion

The results of this study partially support our first hypothesis that there would be a significant difference in the time spent percentage of each of the hip–knee and knee–ankle inter-segmental coordination patterns, and the activity ratio of the agonist and antagonist muscles of the kicking leg between experienced athletes and novices. Specifically, regarding the hip–knee and knee–ankle inter-segmental coordination, the experienced athletes demonstrated a distinct proximal-to-distal pattern across all three kicking phases, characterized by dominant knee flexion during the back swing and leg-cocking phases for energy storage, followed by powerful dominant knee extension during the leg acceleration phase for energy release.

The results of this study showed that the experienced athletes had more hip–knee in-phase, less hip–knee thigh dominance, and more knee–ankle shank dominance coordination patterns compared to the novices in the back swing phase of the kicking movement. Although a statistically significant difference was observed in the time spent percentage of hip–knee in-phase pattern in the back swing phase between the two groups (*p* = 0.05), its practical implication may be very limited due to the small effect size (d = 0.29). Combined with the extension and flexion movement of the hip, knee, and ankle joints, the results (primarily the differences in thigh and shank dominance, rather than the in-phase pattern) indicated that the knee rotated backward dominantly (knee flexion) with a minor hip backward rotation (extension), and ankle forward rotation (dorsiflexion) was the main inter-segmental coordination pattern for the experienced athletes in the back swing phase. The dominant knee flexion coordination pattern observed in experienced players is consistent with the automatization of motor skills through extensive practice. This automatization is characterized by more precise inter-segmental timing and the optimized transfer of mechanical energy from proximal to distal segments, culminating in higher velocity at the foot at impact. This is evident in the previous study, which found that the skilled and intermediate groups demonstrated a more precise temporal sequencing and a shift in the relative contribution between proximal and distal joints, characterized by an increased involvement of the distal segments’ dominance (knee and ankle) compared to the novice group [11]. In addition, literature has demonstrated that a ‘tension arc’ was formed by the kicking side hip over-extension, knee flexion, and trunk rotation to provide initial conditions that increase the lengths of trunk flexors, hip flexors, and quadriceps, notably stretching the bi-articular rectus femoris, which spans the hip and knee, before their contraction at the beginning of instep kicking [22,34]. This dynamic muscle pre-lengthening should generate larger muscle forces based on length–tension relationships of the muscles, increasing the effectiveness of kicking. As revealed by a previous study, a pre-lengthening of 120 to 130% (measured from resting length) leads to maximum muscle tension [35]. A previous study also reported that the velocity of the kicking leg increased significantly when the knee extensor muscles were stretched at the beginning of the kicking movement and then shortened, compared with the kicking movement with only knee concentric contraction [36]. The results of the present study and previous study suggested that the dominant knee flexion in the back swing phase of the experienced athletes is a critical factor for increasing the ball speed by stretching the knee extensor muscles and contributing to the ‘tension arc’.

The results showed that the experienced athletes had less hip–knee in-phase, more knee–ankle shank dominance, less knee–ankle anti-phase, and foot dominance coordination patterns compared to the novices in the leg-cocking phase of the kicking movement. Combined with the extension and flexion movement of the hip, knee, and ankle joints, the results indicated that the knee rotated backward dominantly (knee flexion) with a minor hip forward rotation (flexion), and ankle (plantar flexion) backward rotation was the main inter-segmental coordination pattern for the experienced athletes in the leg-cocking phase. Similarly, a previous study also demonstrated the dominant hip–knee coordination pattern of the shank dominance (knee flexion) before and after the moment of support foot touchdown during the kicking movement for eleven female soccer players recruited from the university soccer team [22]. Further, the dominant knee flexion movement in the leg-cocking phase found in the present study could stretch the knee extensor muscles to contribute to the ‘tension arc’ introduced by Shan and Westerhoff [34].

The results also showed that the experienced athletes had less knee–ankle anti-phase and more shank dominance coordination patterns compared to the novices in the leg acceleration phase of the kicking movement. Combined with the extension and flexion movement of the hip, knee, and ankle joints, the results indicated that the knee rotated forward dominantly (knee extension), with a minor ankle forward rotation (dorsiflexion) as the main inter-segmental coordination pattern for the experienced athletes in the leg acceleration phase. This coordination pattern further contributed to increasing the knee extension torque by releasing the ‘tension arc’ and transferring the energy stored during the back swing and leg-cocking phase. Previous research on the biomechanical analysis of instep kicking demonstrated that the knee extension torque of experienced athletes reaches the peak value before the foot contact with the ball, which is the key to achieving greater foot and ball speed [17]. This result, combined with the results of the previous study, indicated that the dominant knee extension in the leg acceleration phase was critical for a greater ball speed.

The results of this study indicated that the experienced athletes had a more economical and less injury-prone activity ratio of the agonist and antagonist muscles of the kicking leg compared to the novices. On the one hand, the results showed that the activity ratio of the tibialis anterior and gastrocnemius muscles of the experienced athletes was significantly lower than that of the novices. The results indicated that the gastrocnemius activation was relatively high throughout the whole kicking movement, which helped to keep the ankle plantar flexion and reduce other unnecessary muscle activity. The literature demonstrated that elite athletes achieved a longer distance of kicking with less muscle activation before the foot contact with the ball compared with amateur athletes. This result, combined with the results from the literature, indicated that experienced athletes could effectively utilize the muscle system and improve the economic efficiency of kicking movements. On the other hand, the highly activated gastrocnemius muscle also maintained the stiffness of the ankle joint when the foot made contact with the ball, which may reduce the injury risk of the ankle joint.

The results of this study partially support our second hypothesis that the maximum ball speed and the kicking accuracy would be significantly correlated with the time spent percentage of each of the hip–knee and knee–ankle inter-segmental coordination patterns, and the activity ratio of the agonist and antagonist muscles of the kicking leg for the experienced athletes. Specifically, the results of this study showed that a greater time spent percentage of hip–knee shank dominance, as well as the knee–ankle foot dominance coordination pattern in the leg acceleration phase of the kicking movement, was associated with increased maximum ball speed. It should be noted that although a greater time spent percentage of the hip–knee in-phase coordination pattern in the back swing phase was also correlated with ball speed, its practical importance is likely limited due to the small effect size for its group difference (d = 0.29). Instead, a lower time spent percentage of the hip–knee thigh dominance and knee–ankle foot dominance coordination patterns in the back swing phase, knee–ankle anti-phase coordination pattern in the leg-cocking phase, and hip–knee in-phase and thigh dominance coordination patterns in the leg acceleration phase of the kicking movement was associated with increased maximum ball speed. Combined with the extension and flexion movement of the hip, knee, and ankle joints, these results indicated that the athletes achieving greater maximum ball speed had the characterization of inter-segmental coordination, where the knee flexion was dominant with a minor hip extension in the back swing phase, and the knee extension and ankle plantar flexion were dominant in the leg acceleration phase of the kicking movement. As discussed above, the dominant knee flexion in the back swing phase was important for stretching the knee extensor muscles to contribute to the ‘tension arc’, while the dominant knee extension in the leg acceleration was critical for the release of the ‘tension arc’ by transferring the energy stored during the back swing and finally increasing the ball speed. The dominant ankle plantar flexion in the leg acceleration phase helped reduce foot deformation at impact and increase the coefficient of restitution [13]. The results of the current study were also supported by the results of previous studies. The majority of previous studies on soccer kick biomechanics have demonstrated that the ball speed is the result of various factors, in which the foot velocity before the foot contact with the ball, as well as the mechanics of foot-to-ball contact, were the most important [13,17]. The foot velocity before the foot contact with the ball depends on the velocity of the knee extension at that moment due to the importance of the proximal-to-distal sequence of segmental angular velocities and the optimum transfer of energy between segments for kicking performance [13,37,38]. The literature also demonstrated that among high-skilled athletes, those who demonstrate higher knee extension moment during the kick achieve a higher foot velocity [39]. It is worth noting that although several studies suggested that angular velocities of hip flexion, knee extension, and ankle plantar flexion of the kicking leg before the foot contact with the ball were significantly positively correlated with the maximum ball speed [14,15,16], the results of this current study emphasized the importance of the knee extension dominance by examining lower extremity inter-segmental coordination instead of single-joint movement.

The study found that the time spent percentage of the hip–knee thigh dominance and knee–ankle foot dominance coordination patterns in the back swing, as well as the activity ratio of the tibialis anterior and gastrocnemius muscles, were positively correlated with the distance between ball center and goal center. The results indicated that the less hip extension dominance and ankle dorsiflexion dominance in the back swing phase, and the lower the activity ratio of the tibialis anterior and gastrocnemius muscles, the better the kicking accuracy, because the shorter distance between the ball center and goal center suggested a better kicking accuracy in this study. The lower activity ratio of the tibialis anterior and gastrocnemius muscles indicated high activation of the gastrocnemius muscles and low activation of the tibialis anterior muscles, which is conducive to maintaining ankle plantar flexion during the kicking movement. It has been suggested that sources of inaccuracy arise from the error in the force applied by the foot when the foot makes contact with the ball, which depends on the coordinated activation of the muscles precisely controlled by the nervous system [40,41]. These results combined indicated that the precise control of muscle activation by the nervous system is critical for accuracy during instep kicking. In addition, the literature also demonstrated that the biomechanics of the support leg play a significant role in kicking accuracy. The positive knee flexion of the support leg during the landing phase, on the one hand, contributed to reducing the ground reaction force and then maintaining body stability [42]. On the other hand, the knee flexion of the support leg allowed the body sufficient time to adjust the posture according to the ball direction by reducing the approaching velocity [42]. These results indicate that more attention to the effects of the inter-segmental coordination of the support leg on kicking accuracy is needed.

The present study characterized the inter-segmental coordination of the hip, knee, and ankle movements of the kicking leg during instep kicking for experienced athletes and novices using a vector coding method. Further studies are needed to explore other inter-segmental coordination patterns for the support leg, the trunk, and pelvis of the body, because the success of instep kicking is attributed to the coordination of the whole body [16]. Future investigations should include kinetic data and validate these findings under real-game conditions with varied tactical shooting scenarios. Longitudinal studies are also needed to confirm that the inter-segmental coordination patterns can be improved by specific training programs.

It should be noted that the definition of the novice group, which was based on the absence of organized soccer experience rather than standardized skill assessments, may have introduced additional variability in their movement patterns. While this approach effectively captured a population with genuine beginner-level exposure, it represents a limitation in the group’s homogeneity. Further, the relatively small sample size of the experienced group (*n* = 14) may have limited the statistical power of our analyses. Future investigations with larger, balanced cohorts are needed to draw more definitive conclusions. Finally, a fixed cutoff frequency of 13 Hz was used for the low-pass filtering of kinematic data. While this value is well-established in the literature [25,26,27], the use of a universal cutoff without participant-specific residual analysis may, in some instances, lead to over-smoothing or incomplete noise removal. Future research would benefit from employing data-driven methods like residual analysis to determine the optimal cutoff frequency on a trial-by-trial basis.

## 5. Conclusions

In conclusion, this study demonstrates that kicking speed and accuracy are governed by distinct neuromuscular mechanisms. Greater ball speed is primarily driven by a proximal-to-distal inter-segmental coordination pattern dominated by knee flexion (in back swing and leg-cocking) and extension (in leg acceleration). In contrast, kicking accuracy is linked to refined neuromuscular control at the distal ankle. It is important to note that our analysis of accuracy was confined to the kicking leg, and the roles of the support leg and trunk remain critical for a complete understanding. Based on these findings, training recommendations can be precisely targeted. Dynamic stability exercises that reinforce the proximal-to-distal sequence of knee flexion and extension are recommended to improve ball speed, while neuromuscular drills focused on selective activation and relaxation of the ankle musculature are advocated to enhance the task-specific control necessary for accuracy.

## Figures and Tables

**Figure 1 bioengineering-12-01151-f001:**
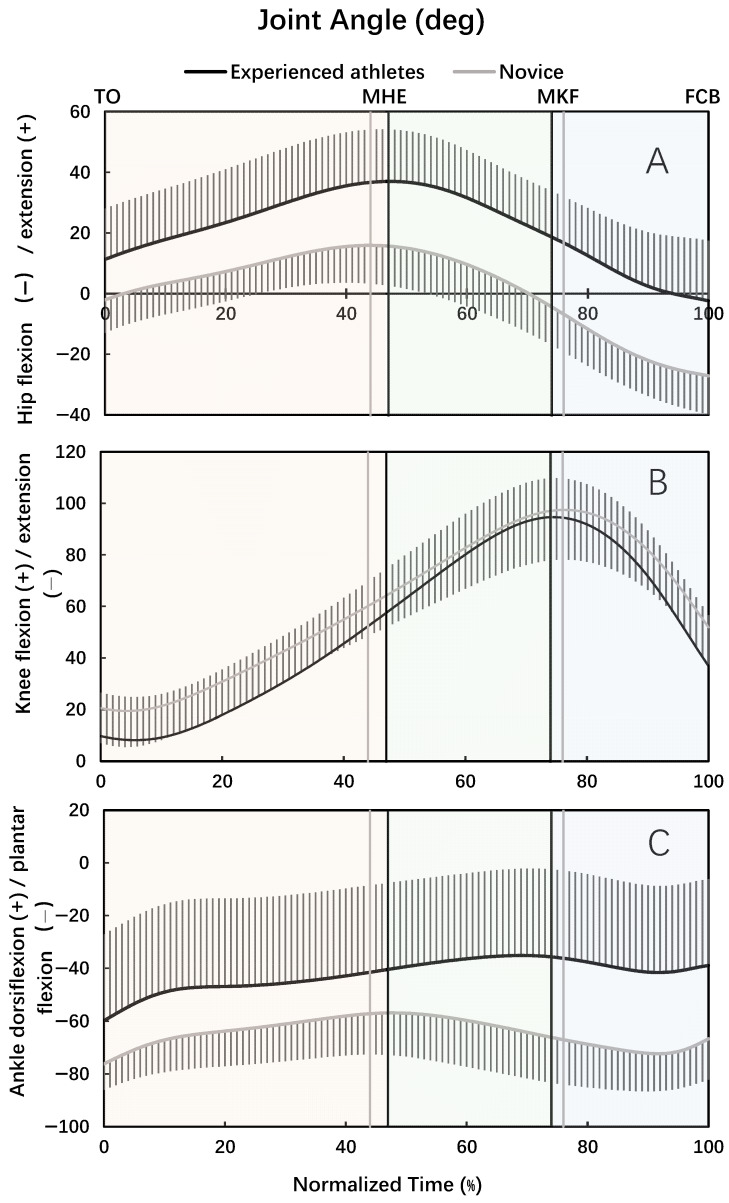
Hip flexion (−)/extension (+) angle (**A**), knee flexion (+)/extension (−) angle (**B**), and ankle dorsiflexion (+)/plantar flexion (−) (**C**) during the kicking movement (TO = toe off; MHE = maximal hip extension; MKF = maximal knee flexion; FCB = foot contact with ball). The three-phase time windows are highlighted with shaded backgrounds for the experienced group: Back swing phase in light orange, leg-cocking phase in light green, and leg acceleration phase in light blue.

**Figure 2 bioengineering-12-01151-f002:**
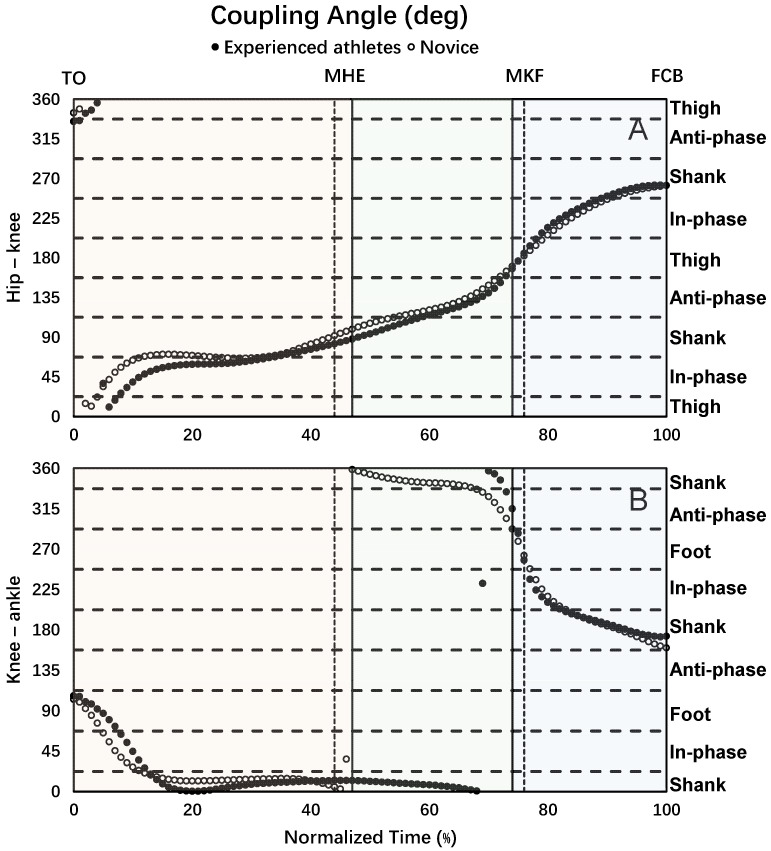
Hip–knee (**A**) and knee–ankle (**B**) coupling angles during the kicking movement (TO = toe off; MHE = maximal hip extension; MKF = maximal knee flexion; FCB = foot contact with ball). The horizontal dash lines defined the coordination patterns for the hip–knee coupling angles: in-phase, anti-phase, thigh dominance (thigh) and shank dominance (shank), and for the knee–ankle coupling angles: in-phase, anti-phase, shank dominance (shank) and foot dominance (foot). The vertical solid lines represent the phase divisions for the experienced group, and the dashed lines represent those for the novice group. The three-phase time windows are highlighted with shaded backgrounds for the experienced group: Back swing phase in light orange, leg-cocking phase in light green, and leg acceleration phase in light blue.

**Figure 3 bioengineering-12-01151-f003:**
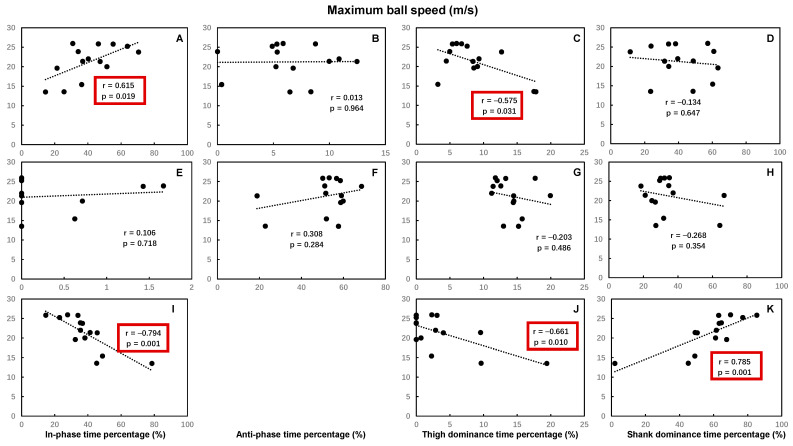
The Pearson correlation coefficient between the maximal ball speed and the time spent percentage of each of the hip–knee coordination patterns during the back swing phase (**A**–**D**), leg-cocking phase (**E**–**H**), and leg acceleration phase (**I**–**K**) of the kicking movement (no coefficient was calculated for the anti-phase since the time spent percentage of anti-phase in leg acceleration phase was 0 for all participants). Statistically significant correlations (*p* < 0.05) are highlighted by red boxes.

**Figure 4 bioengineering-12-01151-f004:**
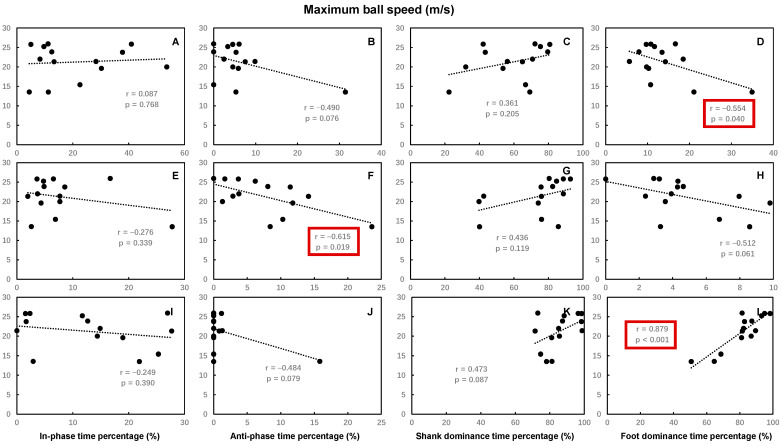
The Pearson correlation coefficient between the maximal ball speed and the time spent percentage of each of the knee–ankle coordination patterns during the back swing phase (**A**–**D**), leg-cocking phase (**E**–**H**), and leg acceleration phase (**I**–**L**) of the kicking movement. Statistically significant correlations (*p* < 0.05) are highlighted by red boxes.

**Figure 5 bioengineering-12-01151-f005:**
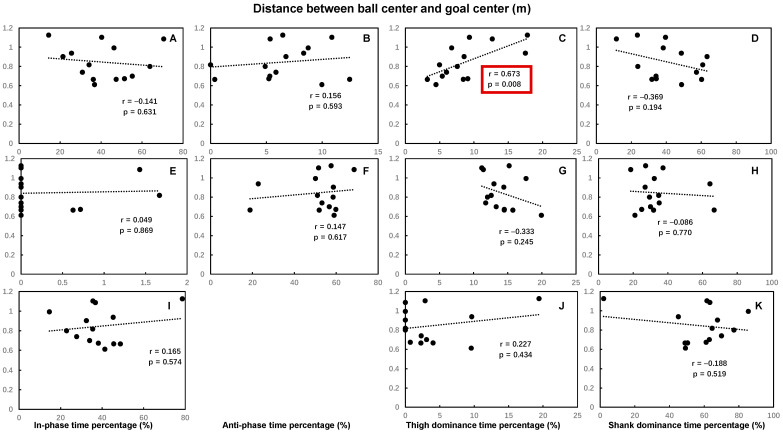
The Pearson correlation coefficient between the distance between the ball center and goal center and the time spent percentage of each of the hip–knee coordination patterns during the back swing phase (**A**–**D**), leg-cocking phase (**E**–**H**), and leg acceleration phase (**I**–**K**) of the kicking movement (no coefficient was calculated for the anti-phase since the time spent percentage of anti-phase in leg acceleration phase was 0 for all participants). Statistically significant correlations (*p* < 0.05) are highlighted by red boxes.

**Figure 6 bioengineering-12-01151-f006:**
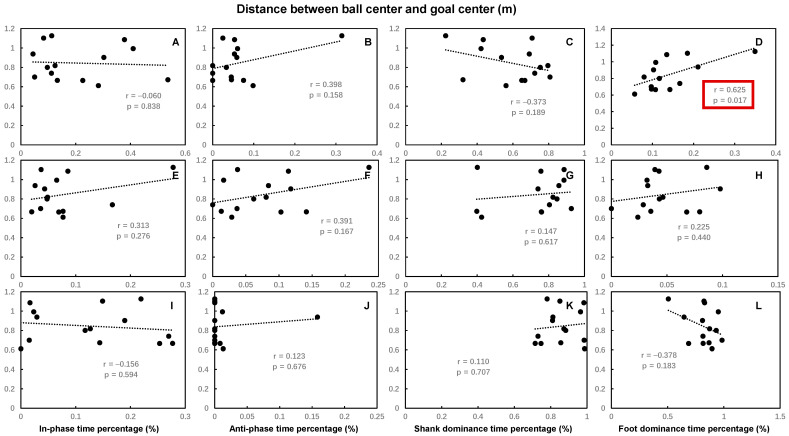
The Pearson correlation coefficient between the distance between the ball center and goal center and the time spent percentage of each of the knee–ankle coordination patterns during the back swing phase (**A**–**D**), leg-cocking phase (**E**–**H**), and leg acceleration phase (**I**–**L**) of the kicking movement. Statistically significant correlations (*p* < 0.05) are highlighted by red boxes.

**Figure 7 bioengineering-12-01151-f007:**
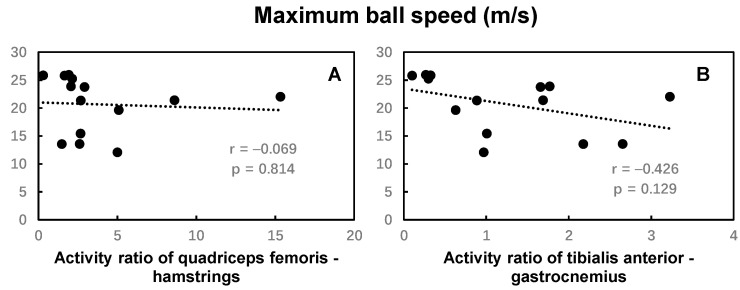
The Pearson correlation coefficient between the maximal ball speed and activity ratio of the quadriceps femoris and hamstrings (**A**), and the tibialis anterior and gastrocnemius muscles (**B**) during the kicking movement.

**Figure 8 bioengineering-12-01151-f008:**
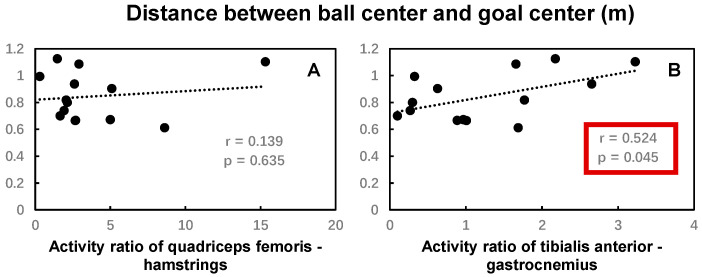
The Pearson correlation coefficient between the distance between ball center and goal center and activity ratio of the quadriceps femoris and hamstrings (**A**), and the tibialis anterior and gastrocnemius muscles (**B**) during the kicking movement. Statistically significant correlations (*p* < 0.05) are highlighted by red boxes.

**Table 1 bioengineering-12-01151-t001:** Scheme used to categorize coordination patterns.

Coordination Pattern	Coupling Angle Definitions
In-phase	22.5°≤γ<67.5°, 202.5°≤γ<247.5°
Anti-phase	112.5° ≤ γ < 157.5°, 292.5° ≤ γ < 337.5°
Proximal dominance	0°≤γ<22.5°, 157.5°≤γ<202.5°, 337.5°≤γ<360°
Distal dominance	67.5°≤γ<112.5°, 247.5°≤γ<292.5°

**Table 2 bioengineering-12-01151-t002:** Comparison of maximal ball speed and distance between ball center and goal center between experienced players and novices.

	Experienced Players (*n* = 14)	Novices (*n* = 32)	*p* Value	Effect Size
Maximal ball speed (m/s)	20.57 ± 5.22	13.22 ± 2.38	**<0.001**	0.67
Distance between ball center and goal center (m)	0.37 ± 0.11	0.99 ± 0.26	**0.031**	0.84

The bold in *p*-Value: Significant difference (*p* < 0.05) between experienced players and novices.

**Table 3 bioengineering-12-01151-t003:** Comparison of time spent percentage of each of the hip–knee coordination patterns in each kicking phase between experienced athletes and novices.

Kicking Phase	Patterns	Experienced Athletes (*n* = 14)	Novices (*n* = 32)	*p* Value	Effect Size
Back swing	In-phase	41.0 ± 15.9	37.3 ± 17.3	**0.050**	0.29
	Anti-phase	6.5 ± 3.5	4.6 ± 2.3	**0.049**	0.30
	Thigh dominance	8.7 ± 4.5	12.6 ± 7.1	**0.032**	0.41
	Shank dominance	41.1 ± 16.0	45.5 ± 19.0	0.631	0.12
Leg-cocking	In-phase	0.3 ± 0.6	0.6 ± 1.0	**0.006**	0.42
	Anti-phase	51.3 ± 13.8	47.1 ± 10.7	0.455	0.17
	Thigh dominance	14.1 ± 2.5	16.2 ± 5.7	0.239	0.23
	Shank dominance	34.3 ± 14.2	35.7 ± 12.3	0.931	0.05
Leg acceleration	In-phase	38.3 ± 14.8	38.7 ± 12.3	0.871	0.01
	Anti-phase	0.0 ± 0.0	0.2 ± 0.7	0.316	0.19
	Thigh dominance	3.8 ± 5.5	6.4 ± 5.6	0.250	0.12
	Shank dominance	57.8 ± 19.6	54.7 ± 15.5	0.762	0.09

The bold in *p*-Value: Significant difference (*p* < 0.05) between experienced players and novices.

**Table 4 bioengineering-12-01151-t004:** Comparison of time spent percentage of each of the knee–ankle coordination patterns in each kicking phase between experienced athletes and novices.

Kicking Phase	Patterns	Experienced Athletes (*n* = 14)	Novices (*n* = 32)	*p* Value	Effect Size
Back swing	In-phase	20.6 ± 15.3	28.3 ± 18.6	0.185	0.22
	Anti-phase	6.2 ± 7.8	6.3 ± 6.8	0.960	0.01
	Shank dominance	59.2 ± 18.2	45.8 ± 23.9	**0.037**	0.41
	Foot dominance	14.0 ± 7.3	19.7 ± 16.9	0.232	0.21
Leg-cocking	In-phase	7.7 ± 6.8	7.1 ± 6.0	0.752	0.05
	Anti-phase	7.7 ± 6.4	35.4 ± 19.2	**<0.001**	0.70
	Shank dominance	73.4 ± 18.5	49.1 ± 21.1	**<0.001**	0.52
	Foot dominance	4.7 ± 2.7	8.4 ± 4.2	**0.004**	0.46
Leg acceleration	In-phase	13.1 ± 10.1	18.1 ± 12.0	0.173	0.20
	Anti-phase	1.4 ± 4.2	8.4 ± 8.9	**0.008**	0.45
	Shank dominance	85.6 ± 9.6	70.2 ± 18.7	**0.006**	0.46
	Foot dominance	0.0 ± 0.0	1.1 ± 2.8	0.053	0.27

The bold in *p*-Value: Significant difference (*p* < 0.05) between experienced players and novices.

**Table 5 bioengineering-12-01151-t005:** Comparison of the average activity ratio of the agonist and antagonist muscles in the kicking movement between experienced players and novices.

	Experienced Athletes (*n* = 14)	Novices (*n* = 32)	*p* Value	Effect Size
Quadriceps femoris–hamstrings	4.10 ± 4.05	6.05 ± 5.53	0.275	0.20
Tibialis anterior–gastrocnemius	1.42 ± 0.94	2.50 ± 1.41	**0.013**	0.44

The bold in *p*-Value: Significant difference (*p* < 0.05) between experienced players and novices.

## Data Availability

Data supporting the results presented in the manuscript are included in the figures and online resources whenever possible and are available upon request to the corresponding author.

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
