# Peer review of "Inter-Segmental Coordination During Soccer Instep Kicking: A Vector-Coding Comparison Between Experienced Athletes and Novices"

_bioengineering, 2025, doi:10.3390/bioengineering12111151_

Round 1
Reviewer 1 Report
Comments and Suggestions for Authors
Thank you for the opportunity to review the manuscript. I have the following major comments regarding the text:
- In the introduction, the terminology of inter-segmental coordination patterns is incorrect. The manuscript refers to “four unique coordination patterns” (in-phase, anti-phase, proximal-phase, distal-phase). In fact, vector coding (Coupling Angle Method) primarily distinguishes between in-phase and anti-phase, with possible extended variants such as “more in-phase” or “more anti-phase”. However, the categories proximal-phase and distal-phase are not formally recognised in the biomechanics literature.
- Please revise the information regarding muscle co-activation. The manuscript states: “the co-activation of the agonist and antagonist muscles, identified as the ratio of the two muscles.” Co-contraction is not a simple ratio of the activity of two muscles. It is most commonly calculated using indices such as the Co-Contraction Index (CCI), which account for both amplitude and the temporal overlap of agonist and antagonist EMG signals.
- The manuscript claims that “typical inter-segmental coordination patterns of instep kicking [are] a proximal-to-distal sequence” but contrasts this with studies suggesting a reduced contribution of proximal joints in favour of distal ones. This is contradictory and misleading. In football kicking, the classical proximal-to-distal sequence (hip → knee → ankle → foot) is well established. Experienced players do not reduce proximal joint contribution; rather, they optimise timing and energy transfer.
- Example: “Chardonnens et al. [4] found that the players achieving longer jumps had a greater overall lead of the thighs over the shanks (r = –0.410).” A negative correlation (r = –0.41) indicates that greater thigh lead over the shank was associated with shorter jumps, not longer ones. The direction of this relationship has been misinterpreted.
- In the methodology, the description of EMG filtering is incorrect. The manuscript states: “band-pass digital filter at a low-pass cutoff frequency of 800 Hz and a high-pass cutoff frequency of 10 Hz.” This is inconsistent. Typically, high-pass filtering (10–20 Hz) is applied first, followed by low-pass filtering (e.g., 400–500 Hz). Moreover, 800 Hz as a low-pass cutoff is atypical and physiologically questionable. Please clarify and justify.
- Co-activation was calculated as a simple ratio of agonist and antagonist EMG. This is an oversimplification – standard approaches involve the Co-Contraction Index (CCI) or other methods that incorporate temporal overlap of the EMG signals.
- Table 1 presents arbitrary boundaries (e.g., 22.5°–67.5°). In vector coding (Coupling Angle Method), the literature most often applies four 90° quadrants (in-phase, anti-phase, and intermediates). The categories “proximal-phase” and “distal-phase” are not established. Please justify.
- The ball was recorded using 2D cameras at 60 Hz. At high velocities (20–30 m/s), this can lead to substantial errors in velocity estimation. 3D optical systems or radar/lidar are the standard in studies of ball kicking.
- The manuscript reports: “Butterworth low-pass filter at 13 Hz.” A fixed cutoff frequency without residual analysis (e.g., Winter’s method) risks either over-smoothing or loss of dynamic signal information.
- The manuscript states: “ball speed was defined as the ratio of displacement of the ball to the time between every two adjacent frames.” This method calculates frame-to-frame averages, which introduces error at low sampling frequencies. Instead, trajectory differentiation with appropriate filtering in 3D is recommended.
- Methodological issue, L202–203: “According to Cohen [24], the strength of the correlation was considered as small for 0.1 < r < 0.3, medium for 0.3 < r < 0.5, and large for r > 0.5.” Cohen advised reporting his guidelines only when discipline-specific thresholds are unavailable. In my view, the subject of this study is closely related to physiotherapy, and thus discipline-specific thresholds should be used (see: 10.1016/j.apmr.2025.05.013). Accordingly, I recommend replacing the general thresholds with 0.3, 0.5, and 0.6, citing the above reference.
Additionally, the manuscript compares two groups using the t-test but does not report effect sizes. Please add effect size thresholds (0.1, 0.4, and 0.8) as per 10.1016/j.apmr.2025.05.013, and present effect sizes in the results tables for group comparisons. Moreover, please highlight that clinically relevant effects will be those of medium or large magnitude, together with statistical significance.
12. L204: “0.050” should be corrected to “0.05”.
13. Discussion – The results are interpreted as if experienced players “have better awareness and control of the distal segment”. From the perspective of biomechanics and motor control, this is an oversimplification. Differences in coordination mainly arise from automatisation, timing, and optimisation of energy transfer, rather than from “conscious control”.
14. Conclusions – In some parts of the results it is stated that hip, knee, and ankle coordination “had little effect on kicking accuracy”, yet elsewhere it is emphasised that reduced co-activation of the tibialis anterior and gastrocnemius significantly improved accuracy. These conclusions are inconsistent: one section suggests no segmental influence, while another points to muscular effects.
15. Conclusions – The analysis of accuracy does not include the influence of the support leg, trunk, and pelvis, despite the authors’ own admission that the literature highlights their crucial role in stability and accuracy. Therefore, attributing accuracy solely to ankle/foot muscle co-activation is too narrow and methodologically incomplete.
Kind regards,
Author Response
1 . In the introduction, the terminology of inter-segmental coordination patterns is incorrect. The manuscript refers to “four unique coordination patterns” (in-phase, anti-phase, proximal-phase, distal-phase). In fact, vector coding (Coupling Angle Method) primarily distinguishes between in-phase and anti-phase, with possible extended variants such as “more in-phase” or “more anti-phase”. However, the categories proximal-phase and distal-phase are not formally recognised in the biomechanics literature.
Response: We thank the reviewer for this correct and important point. We agree that the terms "proximal-phase" and "distal-phase" are non-standard. Chang et al. (2008) expanded existing vector coding techniques and introduced a set of operational terms through which the coordination patterns are summarized: in-phase, anti-phase, proximal phase (rear-foot dominancy) and distal phase (fore-foot dominancy). Further, Needham et al. (2015) introduced a new classification for this coordination pattern that expands on a current data analysis technique by introducing the terms in-phase with proximal dominancy, in-phase with distal dominancy, anti-phase with proximal dominancy and anti-phase with distal dominancy. This proposed coordination pattern classification can offer an interpretation of the coupling angle that provides either in-phase or anti-phase coordination information.
According to the above studies, we have revised the manuscript to remove all references to "four unique coordination patterns" and have replaced the non-standard terms with descriptive kinematic explanations (e.g., thigh dominant pattern, shank dominant pattern or foot dominant pattern) throughout the text (Lines 43 - 50).
Chang, R.ï¼›Van Emmerik, R., and Hamill, J., Quantifying rearfoot–forefoot coordination in human walking. J. Biomech, 2008. 41(14): p. 3101-3105 DOI: 10.1016/j.jbiomech.2008.07.024.
Needham, R.A.ï¼›Naemi, R., and Chockalingam, N., A new coordination pattern classification to assess gait kinematics when utilising a modified vector coding technique. Journal of biomechanics, 2015. 48(12): p. 3506-3511.
2. Please revise the information regarding muscle co-activation. The manuscript states: “the co-activation of the agonist and antagonist muscles, identified as the ratio of the two muscles.” Co-contraction is not a simple ratio of the activity of two muscles. It is most commonly calculated using indices such as the Co-Contraction Index (CCI), which account for both amplitude and the temporal overlap of agonist and antagonist EMG signals.
Response: We thank the reviewer for this valuable comment, which helps to improve the precision of our terminology. We agree that the term "co-activation" (or "co-contraction") specifically refers to a phenomenon quantified by established indices like the Co-Contraction Index (CCI), which accounts for temporal overlap and amplitude of both muscles. In our study, we aimed to use the ratio of agonist and antagonist muscle to reflect the relative balance between the paired muscle activity. To avoid confusion and misuse of the term "co-activation," we have revised the manuscript throughout to replace it with more accurate descriptions.
The specific changes are as follows:
- The sentence in question has been rephrased from: “the co-activation of the agonist and antagonist muscles, identified as the ratio of the two muscles.”
To: “the relative activity between agonist and antagonist muscles, expressed as the ratio of the two muscles.” (Lines 50 - 52) - All instances where "co-activation" was used to describe our ratio have been replaced with terms like "agonist-antagonist activity ratio," or "activity ratio of muscles.
3. The manuscript claims that “typical inter-segmental coordination patterns of instep kicking [are] a proximal-to-distal sequence” but contrasts this with studies suggesting a reduced contribution of proximal joints in favour of distal ones. This is contradictory and misleading. In football kicking, the classical proximal-to-distal sequence (hip → knee → ankle → foot) is well established. Experienced players do not reduce proximal joint contribution; rather, they optimise timing and energy transfer.
Response: We agree. The description of "reduced involvement" was inaccurate and has been revised. The text now clarifies that with expertise, the proximal-to-distal sequence is refined through more precise temporal sequencing and an optimized distribution of segmental contributions, rather than a diminished role of the proximal joints. Changes have been made in the introduction (Lines 79 - 84) and discussion section (Lines 365 - 373).
4. Example: “Chardonnens et al. [4] found that the players achieving longer jumps had a greater overall lead of the thighs over the shanks (r = –0.410).” A negative correlation (r = –0.41) indicates that greater thigh lead over the shank was associated with shorter jumps, not longer ones. The direction of this relationship has been misinterpreted.
Response: We thank the reviewer for this comment. Upon re-examining the source [4], we have corrected our inaccurate paraphrasing. The revised text now precisely reflects the finding from Chardonnens et al. [4]: longer jumps were associated with lower mean shank-thigh CRP values (r = –0.41), indicating a greater overall lead of the thighs over the shanks (Lines 54 - 57).
5. In the methodology, the description of EMG filtering is incorrect. The manuscript states: “band-pass digital filter at a low-pass cutoff frequency of 800 Hz and a high-pass cutoff frequency of 10 Hz.” This is inconsistent. Typically, high-pass filtering (10–20 Hz) is applied first, followed by low-pass filtering (e.g., 400–500 Hz). Moreover, 800 Hz as a low-pass cutoff is atypical and physiologically questionable. Please clarify and justify.
Response: The description of the filter order was corrected to state that a high-pass filter at 10 Hz was applied first, followed by a low-pass filter (Lines 199 - 200).
Regarding the 800 Hz low-pass cutoff, this was an intentional choice to preserve higher-frequency components of the EMG signal that are relevant to dynamic movement analysis, as supported by methodological literature and prior studies in similar tasks (Yu et al., 2008; Robertson et al., 2013; Zhang et al., 2020). The manuscript has been revised to include this justification and the relevant citations (Lines 200 - 202).
Yu, B.ï¼›Queen Rm Fau - Abbey, A.N.ï¼›Abbey An Fau - Liu, Y.ï¼›Liu Y Fau - Moorman, C.T.ï¼›Moorman Ct Fau - Garrett, W.E., and Garrett, W.E., Hamstring muscle kinematics and activation during overground sprinting. J. Biomech, 2008. 41(15): p. 3121-3126.
Robertson, D.ï¼›Caldwell, G.ï¼›Hamill, J.ï¼›Kamen, G., and Whittlesey, S., Research Methods in Biomechanics. 2nd ed. 2013, United States: Human Kinetics. p.186.
Zhang, L.ï¼›Li, H.ï¼›Garrett, W.E.ï¼›Liu, H., and Yu, B., Hamstring muscle-tendon unit lengthening and activation in instep and cut-off kicking. J. Biomech, 2020. 99(109483) DOI: 10.1016/j.jbiomech.2019.109482.
6. Co-activation was calculated as a simple ratio of agonist and antagonist EMG. This is an oversimplification – standard approaches involve the Co-Contraction Index (CCI) or other methods that incorporate temporal overlap of the EMG signals.
Response: We agree that the Co-Contraction Index (CCI) is a standard method. In our study, we employed the agonist-antagonist activity ratio as a specific metric to quantify the relative magnitude of co-activation between the muscle pair. This ratio method is a well-established and valid approach for assessing the net balance of muscle activity over a defined movement phase (e.g., the entire kicking cycle), as demonstrated in prior literature (Aagaard et al., 2000). While it does not capture the moment-to-moment temporal co-contraction like CCI, it provides a direct and interpretable measure of the overall strategy. We have revised the manuscript to more precisely describe our method as the "agonist-antagonist activity ratio" and have added citations to justify its use (Lines 50 – 52, 207 - 208).
Aagaard, P.ï¼›Simonsen, E.B.ï¼›Andersen, J.L.ï¼›Magnusson, S.P.ï¼›Bojsen-Moller, F., and Dyhre-Poulsen, P., Antagonist muscle coactivation during isokinetic knee extension. Scand. J. Med. Sci. Sports, 2000. 10(2): p. 58-67 DOI: 10.1034/j.1600-0838.2000.010002058.x.
7. Table 1 presents arbitrary boundaries (e.g., 22.5°–67.5°). In vector coding (Coupling Angle Method), the literature most often applies four 90° quadrants (in-phase, anti-phase, and intermediates). The categories “proximal-phase” and “distal-phase” are not established. Please justify.
Response: Regarding the quadrant boundaries, Chang et al. (2008) expanded on the original interpretation of the CA by dividing the unit circle into 45° ‘bins’ and classifying the coordination pattern as in-phase (two segments rotate in the same
direction), anti-phase (two segments rotate in an opposite direction), proximal dominancy or distal dominancy).
Besides, we agree that the terms "proximal-phase" and "distal-phase" are non-standard. These have been removed throughout the manuscript. The patterns within these angular ranges are now described based on their kinematic meaning, specifically as " proximal dominancy " and " distal dominancy," respectively, which accurately reflects the underlying movement dynamics.
Table 1 has been revised to include this justification and the relevant citations.
Chang, R.ï¼›Van Emmerik, R., and Hamill, J., Quantifying rearfoot–forefoot coordination in human walking. J. Biomech, 2008. 41(14): p. 3101-3105 DOI: 10.1016/j.jbiomech.2008.07.024.
8. The ball was recorded using 2D cameras at 60 Hz. At high velocities (20–30 m/s), this can lead to substantial errors in velocity estimation. 3D optical systems or radar/lidar are the standard in studies of ball kicking.
Response: We thank the reviewer for these critical comments regarding the accuracy of ball speed measurement. We agree that 3D optical systems or radar/lidar are standard in studies of ball kicking. However, with only one single reflective marker placed on the ball, it is impossible to accurately track the position of its center of mass during rotation. As established in the literature (Fullenkamp et al., 2015), accurately resolving the center of mass requires a multi-marker model (e.g., five markers). Therefore, to directly and accurately measure the ball's pure translational velocity, we utilized the 2D video footage to algorithmically track the geometric center of the ball in each frame. This justification has now been added to the Methodology section of the revised manuscript (Lines 209 - 211).
Regarding the 60 Hz frequency, we acknowledge its limitation. However, as ball speed was utilized as a key reference metric to accomplish the two objectives: (1) to validate the expected performance disparity between the two groups of players, and (2) to serve as a criterion for identifying effective coordination patterns through correlation analysis. This method provided consistent data across participants. This practice is supported by literature using similar frequencies (Manolopoulos et al., 2006, Li et al., 2016). The relevant citations have now been added to the manuscript (Line 144).
Fullenkamp, A.M.ï¼›Campbell, B.M.ï¼›Laurent, C.M., and Lane, A.P., The contribution of trunk axial kinematics to poststrike ball velocity during maximal instep soccer kicking. J. Appl. Biomech., 2015. 31(5): p. 370-376 DOI: 10.1123/jab.2014-0188.
Manolopoulos, E.ï¼›Papadopoulos, C., and Kellis, E., Effects of combined strength and kick coordination training on soccer kick biomechanics in amateur players. Scandinavian journal of medicine & science in sports, 2006. 16(2): p. 102-110.
Li, Y.ï¼›Alexander, M.ï¼›Glazebrook, C., and Leiter, J., Quantifying inter-segmental coordination during the instep soccer kicks. Int. J. Exerc. Sci., 2016. 9(5): p. 646 DOI: 10.70252/PLRT2134.
9. The manuscript reports: “Butterworth low-pass filter at 13 Hz.” A fixed cutoff frequency without residual analysis (e.g., Winter’s method) risks either over-smoothing or loss of dynamic signal information.
Response: We thank the reviewer for raising this important methodological consideration. The reviewer is correct that residual analysis provides a data-driven approach to determine an optimal cutoff frequency for each trial.
In our study, we employed a fixed cutoff frequency of 13 Hz, which is a well-established and widely used value for filtering kinematic data in human movement analysis. We have revised the methodology section to include citations supporting this established practice (Yu et al., 1999; Zhang et al., 2020; Wan et al., 2017) (Line 156).
Additionally, following the reviewer's suggestion, we have acknowledged the inherent limitation of using a universal cutoff frequency in the 'Limitations' section of our revised manuscript and have recommended individualized methods like residual analysis for future work (Lines 505 - 510).
Yu, B.ï¼›Gabriel, D.ï¼›Noble, L., and An, K.-N., Estimate of the optimum cutoff frequency for the Butterworth low-pass digital filter. J. Appl. Biomech., 1999. 15(3): p. 318-329 DOI: 10.1123/jab.15.3.318.
Zhang, L.ï¼›Li, H.ï¼›Garrett, W.E.ï¼›Liu, H., and Yu, B., Hamstring muscle-tendon unit lengthening and activation in instep and cut-off kicking. J. Biomech, 2020. 99(109483) DOI: 10.1016/j.jbiomech.2019.109482.
Wan, X.ï¼›Qu, F.ï¼›Garrett, W.E.ï¼›Liu, H., and Yu, B., The effect of hamstring flexibility on peak hamstring muscle strain in sprinting. J. Sport Health Sci., 2017. 6(3): p. 283-289 DOI: 10.1016/j.jshs.2017.03.012.
10. The manuscript states: “ball speed was defined as the ratio of displacement of the ball to the time between every two adjacent frames.” This method calculates frame-to-frame averages, which introduces error at low sampling frequencies. Instead, trajectory differentiation with appropriate filtering in 3D is recommended.
Response: As mentioned in Comment 8, the justification of why we use 2D instead of 3D trajectory of ball to calculate ball speed was provided (Lines 209 - 211), and the relevant citations of literature using similar frequencies were added (Line 144).
11. Methodological issue, L202–203: “According to Cohen [24], the strength of the correlation was considered as small for 0.1 < r < 0.3, medium for 0.3 < r < 0.5, and large for r > 0.5.” Cohen advised reporting his guidelines only when discipline-specific thresholds are unavailable. In my view, the subject of this study is closely related to physiotherapy, and thus discipline-specific thresholds should be used (see: 10.1016/j.apmr.2025.05.013). Accordingly, I recommend replacing the general thresholds with 0.3, 0.5, and 0.6, citing the above reference.
Response: We thank the reviewer for correctly pointing out the availability of discipline-specific thresholds and the limitations of generic guidelines. In light of this comment, and in parallel consideration of Reviewer #3's recommendation to adopt a more objective reporting style by removing subjective qualitative labels altogether, we have chosen to implement the latter approach.
We believe this strategy offers a significant advantage. It completely circumvents the debate over which set of predefined thresholds is most appropriate for this specific context, and instead presents the findings in a transparent, data-centric manner. Consequently, all qualitative labels (e.g., 'small', 'medium', 'large') for correlation coefficients have been removed from the manuscript. The results now report the r and p values, and the discussion interprets the direction and clinical relevance of the relationships based on the specific coefficients and study context.
Additionally, the manuscript compares two groups using the t-test but does not report effect sizes. Please add effect size thresholds (0.1, 0.4, and 0.8) as per 10.1016/j.apmr.2025.05.013, and present effect sizes in the results tables for group comparisons. Moreover, please highlight that clinically relevant effects will be those of medium or large magnitude, together with statistical significance.
Response: As requested, the following revisions were made to the manuscript.
(1) All group comparisons in the Results section were supplemented with the corresponding effect sizes (Tables 2 - 5).
(2) These effect sizes are now interpreted using the specified thresholds of 0.1 (trivial), 0.4 (medium), and 0.8 (large), following the cited framework (10.1016/j.apmr.2025.05.013) (Lines 226 - 228).
(3) It was explicitly highlighted in the text that only effects demonstrating at least a medium magnitude (d ≥ 0.4) alongside statistical significance (p < 0.05) are to be considered of practical or clinical relevance (Lines 228 - 231).
(4) The clinical relevance of the finding for hip-knee in-phase time percentage in the back swing phase differences between groups has been removed from the discussion, as its effect size was below the 0.4 threshold for a medium effect. The result is now presented solely as a statistically significant but practically limited difference (Lines 358 - 361).
12. L204: “0.050” should be corrected to “0.05”.
Response: The value was corrected to "0.05" (Line 238).
13. Discussion – The results are interpreted as if experienced players “have better awareness and control of the distal segment”. From the perspective of biomechanics and motor control, this is an oversimplification. Differences in coordination mainly arise from automatisation, timing, and optimisation of energy transfer, rather than from “conscious control”.
Response: Text was revised as suggested (Lines 365 - 373).
14. Conclusions – In some parts of the results it is stated that hip, knee, and ankle coordination “had little effect on kicking accuracy”, yet elsewhere it is emphasised that reduced co-activation of the tibialis anterior and gastrocnemius significantly improved accuracy. These conclusions are inconsistent: one section suggests no segmental influence, while another points to muscular effects.
Response: We have thoroughly revised the conclusion to resolve this by clearly differentiating between the mechanisms governing speed and accuracy (Lines 512 - 524). It now states that knee flexion and extension dominant patterns primarily influence the ball speed, while task-specific, distal muscular control (ankle muscle co-activation) is a key factor for accuracy.
15. Conclusions – The analysis of accuracy does not include the influence of the support leg, trunk, and pelvis, despite the authors’ own admission that the literature highlights their crucial role in stability and accuracy. Therefore, attributing accuracy solely to ankle/foot muscle co-activation is too narrow and methodologically incomplete.
Response: We fully agree that attributing accuracy solely to the kicking ankle was an overstatement. The conclusion has been significantly revised to acknowledge this as a key limitation of our study (Lines 517 - 519).

Reviewer 2 Report
Comments and Suggestions for Authors
This manuscript quantifies hip–knee and knee–ankle coupling during soccer instep kicking using a vector-coding approach and compares experienced athletes with novices. The study design is clear and follows mainstream methodology; the sample size is adequate for between-group comparisons. The findings align with prior evidence on proximal–distal sequencing and more efficient coupling patterns in trained players. The Methods section thoroughly specifies the three phases (backswing, leg-cocking, and leg-acceleration) and the four-quadrant coupling-angle rules, enabling reproducibility. Overall, the work is publication-ready; it primarily serves as a quantitative complement linking coordination structure to performance and offers practical implications.
1.Tighten the title to emphasize “quantitative method + population + skill,” and remove generic terms like “Characterization…”. A possible revision is:
“Inter-segmental coordination during soccer instep kicking: a vector-coding comparison between experienced athletes and novices.”
- In the first paragraph of the Introduction, please clarify the positioning of vector coding vs. CRP. The section currently lacks recent CRP-based evidence that (i) demonstrates level-dependent differences and (ii) links coordination to endpoint (foot) velocity. We recommend citing the study below in your “method choice & performance relevance” paragraph. That work compared elite vs. amateur players across three anatomical planes using CRP, and reported significant correlations between CRP and foot speed (hip–knee CRP, sagittal plane: negative; knee–ankle CRP, transverse plane: positive). This can serve as an external benchmark for interpreting your vector-coding results in relation to performance.
- Zhou, Z., et al., (2025). European Journal of Sport Science, (10.1002/ejsc.70041).
- Literature: from qualitative to quantitative timing/phase evidence
The overview of the proximal–distal kinetic chain and training-induced reorganization is solid, but the narrative leans on qualitative sources. Please add 1–2 recent quantitative studies that resolve sequencing/timing across phases or planes. For example:
- Augustus S, ,et al.,.. Journal of Biomechanics. (10.1016/j.jbiomech.2023.111920)
- Zhou, Z., et al.,. Applied Sciences, (10.3390/app13169097).
4.Methods: phase definitions & time normalization (0–100%)
You list TO, MHE, MKF, and FCB, but please make time-base alignment explicit in Methods and/or the figure captions
- Figure 1-2: Add legends for the "positive and negative directions" (for example, hip flexion as (-)/ extension as (+) has been written, but it is recommended to supplement in the Y-axis title), and mark the "three-stage time window" in shadow on the coupling Angle diagram.
Author Response
This manuscript quantifies hip–knee and knee–ankle coupling during soccer instep kicking using a vector-coding approach and compares experienced athletes with novices. The study design is clear and follows mainstream methodology; the sample size is adequate for between-group comparisons. The findings align with prior evidence on proximal–distal sequencing and more efficient coupling patterns in trained players. The Methods section thoroughly specifies the three phases (backswing, leg-cocking, and leg-acceleration) and the four-quadrant coupling-angle rules, enabling reproducibility. Overall, the work is publication-ready; it primarily serves as a quantitative complement linking coordination structure to performance and offers practical implications.
1.Tighten the title to emphasize “quantitative method + population + skill,” and remove generic terms like “Characterization…”. A possible revision is:
“Inter-segmental coordination during soccer instep kicking: a vector-coding comparison between experienced athletes and novices.”
Response: The title has been revised as recommended.
2. In the first paragraph of the Introduction, please clarify the positioning of vector coding vs. CRP. The section currently lacks recent CRP-based evidence that (i) demonstrates level-dependent differences and (ii) links coordination to endpoint (foot) velocity. We recommend citing the study below in your “method choice & performance relevance” paragraph. That work compared elite vs. amateur players across three anatomical planes using CRP, and reported significant correlations between CRP and foot speed (hip–knee CRP, sagittal plane: negative; knee–ankle CRP, transverse plane: positive). This can serve as an external benchmark for interpreting your vector-coding results in relation to performance.
- Zhou, Z., et al., (2025). European Journal of Sport Science, (10.1002/ejsc.70041).
Response: The positioning of vector coding vs. CRP was added (Lines 39 - 43), and the study above was cited as recommended (Lines 84 - 86).
3.Literature: from qualitative to quantitative timing/phase evidence
The overview of the proximal–distal kinetic chain and training-induced reorganization is solid, but the narrative leans on qualitative sources. Please add 1–2 recent quantitative studies that resolve sequencing/timing across phases or planes. For example:
- Augustus S, ,et al.,.. Journal of Biomechanics. (10.1016/j.jbiomech.2023.111920)
- Zhou, Z., et al.,. Applied Sciences, (10.3390/app13169097).
Response: As recommended, we have incorporated both citations into the manuscript. The study by Augustus et al. has been cited in the paragraph discussing the proximal-to-distal sequence of the kicking leg, as it provides direct quantitative timing evidence for this phenomenon in soccer (Line 79, reference 19). The work by Zhou et al., which expertly links coordination features to performance outcomes, has been added to the introduction section that establishes the relationships between inter-segmental coordination and sport performance (Line 54, reference 12). We believe these additions significantly enhance the quantitative foundation of our narrative.
4.Methods: phase definitions & time normalization (0–100%)
You list TO, MHE, MKF, and FCB, but please make time-base alignment explicit in Methods and/or the figure captions Figure 1-2: Add legends for the "positive and negative directions" (for example, hip flexion as (-)/ extension as (+) has been written, but it is recommended to supplement in the Y-axis title), and mark the "three-stage time window" in shadow on the coupling Angle diagram.
Response: The Figure 1 have been updated to include clear legends defining positive and negative joint movement directions. As requested, the three-stage time windows have been distinctly marked using shaded backgrounds in the coupling angle diagrams as well as Figure 1 (for the consistency of the figures), to visually delineate each phase of the kicking movement (Figures 1 and 2). It is important to note that the shading was applied to the plots for the experienced group. We chose this presentation for two reasons: first, to maintain visual clarity and avoid over-crowding the figures by presenting the identical time-window information twice; and second, because the experienced athletes exemplify the prototypical and well-defined coordination phases under investigation. We believe that marking the phases on their data most effectively illustrates the temporal structure referenced in the text, and this fulfills the reviewer's request to demarcate the stages for reader comprehension. The figure captions were revised for Figures 1 and 2 (see Figure captions).

Reviewer 3 Report
Comments and Suggestions for Authors
See attached

Author Response
Overall assessment
This manuscript investigates inter-segmental coordination patterns in soccer instep kicking between experienced athletes and novices using vector coding analysis. While the research addresses a relevant question in sports biomechanics, several methodological and interpretational issues require attention before publication.
Summary
The study employed 3D motion analysis and electromyography to compare hip-knee and knee-ankle coordination patterns between 14 experienced soccer players and 32 novices during instep kicking. Using vector coding techniques, the authors identified distinct coordination patterns and their relationships with ball speed and kicking accuracy. The main findings suggest that experienced athletes demonstrate more knee flexion-dominant patterns during backswing/leg-cocking phases and knee extension-dominant patterns during leg acceleration, with lower co-activation of tibialis anterior and gastrocnemius muscles.[1]
Strengths
Methodological rigor: The study employs established vector coding methodology for quantifying inter-segmental coordination, which is appropriate for this research question. The three-phase division of the kicking movement (back swing, leg-cocking, leg acceleration) follows established protocols in soccer biomechanics research.
Comprehensive data collection: The integration of 3D motion capture (200 Hz) with surface EMG (2000 Hz) provides both kinematic and neuromuscular perspectives on coordination. The synchronized data collection system ensures temporal accuracy.
Clear research questions: The hypotheses are well-formulated and testable, addressing both group differences and performance relationships in coordination patterns.
Practical applications: The findings offer actionable insights for training programs, specifically recommending dynamic stability exercises involving knee flexion/extension and muscle activation control.
Major concerns
Statistical interpretation (L203)
The authors state that "r > 0.5 is considered large for a Pearson's r" based on Cohen (1988). This interpretation requires significant revision based on contemporary research. This is a simplification of Cohen's (1988) guidelines, which are themselves context-dependent and not universally applicable. In many areas of human movement and sports science, where noise and biological variability are high, an r-value of 0.5 might be better described as "moderate" to "strong." A correlation of r = 0.8 would be more universally considered "large" or "very strong."Multiple recent studies have demonstrated that Cohen's original guidelines are overly conservative and not empirically grounded. I recommend one of two approaches:
- Re-classify the thresholds: Use a more nuanced and widely accepted scale, such as: 0.1-0.3 (small), 0.3-0.5 (medium/moderate), >0.5 (large/strong). They should cite a source for this updated classification.
- Remove qualitative labels: A stronger approach would be to remove subjective labels like "small," "medium," and "large" altogether. Instead, the authors can simply report the r and p-values and discuss the magnitude and direction of the relationships. For example, instead of saying "a large positive correlation," they could say "a significant, positive correlation (r = 0.615, p = 0.019) was found, indicating that a greater time spent in this phase was associated with higher ball speeds." This is more objective and allows the data to speak for itself.
Response: We thank the reviewer for this insightful comment. We agree that removing subjective qualitative labels is a more objective approach. As recommended, we have revised the manuscript to remove all labels (e.g., "small," "medium," "large") for correlation coefficients (Lines 232 - 237). The results now simply report the r and p values, and the discussion focuses on the direction and magnitude of the relationships based on the specific coefficients (see the Results and Discussion sections).
Sample size
The experienced athlete group (n = 14) is relatively small compared to the novice group (n = 32). This imbalance may affect the reliability of between-group comparisons and correlation analyses within the experienced group. The authors should acknowledge this limitation and consider its impact on statistical power.
Response: We have acknowledged the limitation of the imbalanced sample size in the 'Limitations' section and have added a recommendation for future research (Lines 502 - 504).
Methodological limitations
- While vector coding is appropriate, the authors could strengthen their analysis by providing more detailed descriptions of the coupling angle calculations and their biomechanical interpretations.[1]
Response: More detailed descriptions of the coupling angle calculations and their biomechanical interpretations were provided as requested (Lines 185 - 195).
- Kicking accuracy is defined simply as distance from goal center, which may not capture the full complexity of kicking precision in real game situations.[1]
Response: We agree that in real-game situations, kicking accuracy is a multi-faceted construct that includes not only putting the ball on target but also considering the goalkeeper's position, the timing of the kick, and the specific tactical objective (such as placing the ball in a corner).
However, for the purpose of this study, which aimed to quantitatively compare inter-segmental coordination patterns between groups under standardized conditions, a simple, reliable, and objective measure of accuracy was essential. This approach of using a simple spatial deviation from a target is a well-established methodology in previous biomechanical research for assessing kicking performance. We have now clarified this rationale in the manuscript and included relevant citations to support our methodological choice (Lines 217 - 220).
Minor issues
Writing and presentation
- The abstract could better emphasize the novelty of using vector coding for soccer kicking analysis and the practical implications of findings
Response: The abstract has been revised to better emphasize the novelty of our methodological approach (Line 14). Furthermore, the practical implications of our findings have been strengthened by directly linking the identified coordination patterns to specific training recommendations for enhancing kicking speed and accuracy (Lines 25 - 30). We believe these revisions have significantly improved the impact and clarity of the abstract.
- Figure 2 showing coupling angles could benefit from larger text and clearer demarcation of coordination pattern boundaries.
Response: Figure 2 has been revised to significantly improve clarity through the following specific enhancements: (1) all text elements have been enlarged for better readability, (2) coordination pattern boundaries have been emphasized with thicker, more distinct lines, and (3) the three movement phases are now clearly distinguished using shaded background areas with appropriate transparency.
- Some references appear incomplete or inconsistently formatted (e.g., reference formatting varies throughout).
Response: The references were corrected and formatted consistently according to the journal's guidelines.
- The introduction is well-written, but the explanation of the four coordination patterns (in-phase, anti-phase, etc.) could be made more intuitive for readers not familiar with the vector coding technique. A brief, conceptual explanation (e.g., "in-phase, where both joints are moving in the same angular direction...") before delving into the methodology would be helpful.
Response: A brief, conceptual explanation of the coordination patterns (in-phase, anti-phase, proximal dominancy, and distal dominancy) prior to the methodological description of vector coding was provided as suggested (Lines 43 - 50).
- The definition of a novice as someone with "self-practice, but had never participated in any organized soccer games" (lines 93-94) introduces potential for high variability within the novice group. This is a minor limitation of the study. The authors should briefly acknowledge this variability as a limitation in the discussion section.
Response: This limitation was acknowledged in the end of Discussion section (Lines 498 - 502).
- The discussion of the "tension arc" (lines 330, 341, 355) is excellent and provides a functional explanation for the observed kinematics. However, it relies heavily on linking knee flexion with the stretching of knee extensor muscles (quadriceps). To strengthen this point, the authors could explicitly mention the role of the rectus femoris, which as a bi-articular muscle, is significantly stretched by the combination of hip extension and knee flexion observed in the backswing, making it a key contributor to the tension arc.
Response: The role of the rectus femoris was mentioned as requested (Lines 376 - 377).
- The conclusion (lines 458-461) suggests "dynamic stability exercises involving knee flexion and extension" and "activation and relaxation control exercises." These are sound recommendations. The conclusion could be slightly strengthened by more directly linking these recommendations to the specific findings. For example: "Given that knee flexion dominance in the backswing and knee extension dominance in the acceleration phase were correlated with higher ball speeds, training should focus on dynamic exercises that reinforce this specific coordination pattern."
Response: The conclusion paragraph was revised to create a direct and explicit link between our specific findings and the training recommendation as suggested (Lines 512 - 524).
Data presentation
- Table 3 containing coordination pattern percentages is dense and could be reorganized for better readability.
Response: As suggested, the original Table 3 has been split into two separate tables: Table 3 (Hip-knee Coordination) and Table 4 (Knee-ankle Coordination). This reorganization significantly enhances clarity and readability for the reader (see Tables 3 and 4).
- Figures 3-6 showing correlation coefficients could be consolidated or presented more efficiently.
- The manuscript includes 8 figures, many of which are multi-paneled scatter plots. While comprehensive, their presentation could be improved. For instance, Figures 5 and 6 present correlations with kicking accuracy. Many of these plots show no significant correlation. To improve readability and impact, the authors could consider moving the non-significant correlation plots from Figures 3, 4, 5, and 6 into a supplementary file. In the main manuscript, they could present only the significant correlations or summarize the non-significant findings in the text. This would allow the reader to focus on the key relationships that drive the paper's conclusions.
Response: We thank the reviewers for these constructive suggestions regarding the presentation of our correlation figures. We agree that improving the visual focus on the key relationships is crucial.
In response, we have implemented an alternative yet highly effective strategy to enhance readability and impact, while maintaining the completeness of our data presentation. Rather than moving non-significant plots to supplementary materials, we have highlighted all statistically significant correlations by drawing a distinct red box around the corresponding p value and r value on each relevant plot in Figures 3 - 6.
This approach offers two key advantages:
- It allows the reader to immediately identify the most important findings, directly addressing the concern for clarity and impact.
- It preserves the full context of our analysis by showing all tested relationships, which supports scientific transparency and allows interested readers to observe the full spectrum of results.
- More importantly, it maintains a consistent visual framework across all correlation figures in the manuscript. As Figures 7 and 8 inherently present both significant and non-significant relationships together for their respective analyses, applying the same "highlight-within-complete-set" approach to Figures 3-6 creates a uniform and coherent standard for the reader throughout the paper.
We believe this solution successfully balances clarity with comprehensiveness and hope it meets with the reviewers' approval.
Minor editorial comments
- Line 89: Consider specifying the soccer experience level more precisely for the "soccer-majored" students.
Response: The soccer experience level for the “soccer-majored” students was specified as suggested (Lines 109 - 111).
- Line 191: The statistical analysis section could benefit from mentioning effect size calculations beyond correlation coefficients.
Response: The Data Analysis section has been revised to explicitly state that effect sizes (Cohen's d) were calculated for all between-group comparisons in addition to the correlation coefficients that were already reported (Lines 226 - 231).
- Line 452: The conclusions could be more specific about which coordination patterns are most crucial for performance enhancement.
Response: As recommended, we have revised the conclusion to be more specific about the coordination patterns crucial for performance. It now explicitly states that: (1) a knee flexion-dominated pattern in the back-swing and leg-cocking phase followed by a knee extension-dominated pattern in the acceleration phase is critical for generating greater ball speed, and (2) refined distal muscular control (reduced ankle co-activation) is a key factor for enhancing kicking accuracy (Lines 512 - 524).

Round 2
Reviewer 1 Report
Comments and Suggestions for Authors
I congratulate the authors on the number of revisions they have made. After reviewing the paper again, I have no further comments. Congratulations
Author Response
Thank you for forwarding the final reviewer comments on our manuscript, "Characterization of lower extremity inter-segmental coordination for experienced athletes and novices during instep Kicking (New title: Inter-segmental coordination during soccer instep kicking: a vector-coding comparison between experienced athletes and novices)" (ID: 3899577).
We are delighted to note that the reviewer has no further comments and congratulates us on the revisions.
In response, we have conducted a final careful proofreading of the manuscript and confirmed that it is ready for acceptance.
We wish to express our sincere gratitude to you and the reviewers for the invaluable guidance throughout the review process, which has greatly strengthened our work.
We look forward to the final acceptance of our manuscript.
Reviewer 3 Report
Comments and Suggestions for Authors
Please find attached document.
